# BALANCING AVERAGE AND WORST-CASE ACCURACY IN MULTITASK LEARNING

## ABSTRACT

When training and evaluating machine learning models on a large number of tasks, it is important to not only look at average task accuracy—which may be biased by easy or redundant tasks—but also worst-case accuracy (i.e. the performance on the task with the lowest accuracy). In this work, we show how to use techniques from the distributionally robust optimization (DRO) literature to improve worst-case performance in multitask learning. We highlight several failure cases of DRO when applied off-the-shelf and present an improved method, Lookahead-DRO (L-DRO), which mitigates these issues. The core idea of L-DRO is to anticipate the interaction between tasks during training in order to choose a dynamic re-weighting of the various task losses, which will (i) lead to minimal worst-case loss and (ii) train on as many tasks as possible. After demonstrating the efficacy of L-DRO on a small controlled synthetic setting, we evaluate it on two realistic benchmarks: a multitask version of the CIFAR-100 image classification dataset and a large-scale multilingual language modeling experiment. Our empirical results show that L-DRO achieves a better trade-off between average and worst-case accuracy with little computational overhead compared to several strong baselines.

## 1 INTRODUCTION

Multitask learning—the process by which a single model is trained to perform a variety of different tasks—has become a subject of increasing interest with many successful applications in a variety of domains (Ruder, 2017; Yu et al., 2020; Wang et al., 2021). By and large, multitask learning models are evaluated by reporting their average performance on all tasks (McCann et al., 2018; Wang et al., 2019). However, average accuracy does not paint the full picture of a multitask model's performance. Ensuring comparable accuracy across tasks, including on tasks from under-represented training data distributions, is an important issue from both practical and fairness perspectives. For example, in natural language processing, multilingual models often performs worse on languages with smaller amounts of resources on the web (e.g., Wikipedia) due to the resulting scarcity of relevant training data (Hu et al., 2020). In computer vision, facial recognition models have been shown to perform worse on racial groups that are underrepresented in the training data (Buolamwini & Gebru, 2018).

In this paper, we argue that multitask models should also be evaluated by their worst-case accuracy. Yet commonly used task re-weighting based objectives are inadequate to ensure good worst-case performance. Rather, a more natural choice is to use a type of min-max objective from the distributionally robust optimization (DRO) literature (Rahimian & Mehrotra, 2019; Sagawa et al., 2020): $\min_{\boldsymbol{\theta}} \max_i \ell_i(\boldsymbol{\theta})$, where $\ell_i$ is the loss of the $i$-th task and $\boldsymbol{\theta}$ are the model parameters. Training with this objective guarantees at least a certain level of performance on all tasks. However, we observe that DRO does not always work well in practice. Since the min-max objective optimizes the worst loss, it tends to do so by sacrificing performance on other tasks, which hurts average accuracy (§2.3).

Motivated by this observation, we propose a novel optimization algorithm called Lookahead-DRO (L-DRO). The intuition behind L-DRO is to anticipate the evolution of the loss of each task depending on which task we train on. Using this information, L-DRO chooses a weight allocation that (i) leads to minimal worst-case loss and (ii) trains on the maximal number of tasks possible (§2.4).

We evaluate L-DRO on a synthetic task, an image classification task, and a language modeling task. In experiments on a multitask version of the CIFAR-100 dataset, we show that L-DRO consistently allows us to train models that achieve a good trade-off of worst and average task accuracy. On a

realistic large-scale multilingual language modeling dataset, we find that L-DRO is able to improve performance on the worst-performing languages at very little cost to average performance.

Our main contributions are as follows: (i) we show how distributionally robust optimization (DRO) can be applied to multitask learning to maximize worst-case accuracy; (ii) we highlight failure cases of the standard DRO formulation in the multitask setting; (iii) we propose a new algorithm, L-DRO that addresses DRO's deficiencies; and (iv) we demonstrate that L-DRO achieves a better trade-off between average and worst-case accuracy than strong baselines in a synthetic and two realistic experiments from two different modalities.

## 2 TRAINING MULTITASK MODELS

Consider a machine learning model parameterized by $\boldsymbol{\theta} \in \mathbb{R}^{d_{\text{model}}}$, which is trained to perform $N$ tasks. In general, let a task refer to the combination of examples drawn from a data distribution and an objective function. For simplicity, we define a task solely by its objective function $\ell_{i=1...N}$. As a running example, we consider a multilingual language model with dozens of tasks where each task corresponds to a language (Conneau et al., 2020; Xue et al., 2021).

### 2.1 AVERAGE TASK LOSS

The most common objective for multitask learning is to minimize the average of all the task losses:

$$\min_{\boldsymbol{\theta}} \frac{1}{N} \sum_{i=1}^{N} \ell_i(\boldsymbol{\theta})$$

A downside of this objective is that all tasks are treated equally. In real-world scenarios, we often encounter a scenario where task difficulty is not uniform (i.e., the magnitude of $\ell_i$ varies wildly; Hessel et al. 2019) or we want to pay more attention to certain tasks. If the distribution of all tasks is not homogeneous, training with $\mathcal{L}_{\text{avg}}$ results in a model with performance that is skewed towards certain clusters of tasks. In our multilingual example, if a large number of languages come from the same language family such as Indo-European languages, $\mathcal{L}_{\text{avg}}$ will choose models which perform better on these languages at the expense of typologically distinct languages such as Mandarin or Finnish.

### 2.2 STATIC MIXING

Rather than simple averaging, prior work in multitask learning often uses arbitrary weightings $\mathbf{w} = \{w_1, \ldots, w_N\}$ in the probablility simplex $\Delta_N = \{\mathbf{w} \mid w_i > 0, \sum_i w_i = 1\}$ to modulate the losses of individual tasks:

$$\min_{\boldsymbol{\theta}} \sum_{i=1}^{N} w_i \, \ell_i(\boldsymbol{\theta}).$$

The weights are typically decided based on some statistics of the data. In proportional sampling (Sanh et al., 2019), the weight of a task $i$ is proportional to the size of its data $w_i \propto |D_i|$. This biases the model towards tasks that are over-represented in the data. To counter-balance this, recent approaches use an inverse temperature $\alpha$ where $w_i \propto |D_i|^{\alpha}$ with $\alpha$ typically in the range of $[0.2, 0.3]$ (Aharoni et al., 2019; Conneau et al., 2020; Xue et al., 2021).

### 2.3 WORST TASK LOSS

A third option that considers the task on which the model performs worst is the min-max objective. We refer to this objective as DRO because of its prominence in the distributional robustness literature (Sagawa et al., 2020):

$$\min_{\boldsymbol{\theta}} \max_{i} \ell_i(\theta) = \min_{\boldsymbol{\theta}} \underbrace{\max_{\mathbf{w} \in \Delta_N} \sum_{i=1}^{N} w_i \ell_i(\theta)}_{\mathcal{L}_{\text{DRO}}(\boldsymbol{\theta})}.$$

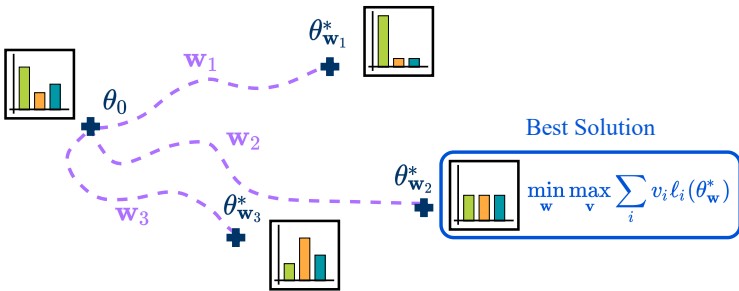

Figure 1: Overview of the Lookahead-DRO procedure. Starting from parameters $\boldsymbol{\theta}_0$, the optimal training weights $\mathbf{w}$ are chosen such that the resulting parameters $\boldsymbol{\theta}_{\mathbf{w}}^*$ minimize the worst case loss.

The above objective minimizes the worst-case loss of the model at every step. Alternatively, we can view the objective similar to the previous ones as minimizing the loss under a certain task weighting $\mathbf{w}$. In this case, the weights $\mathbf{w}$ are not static, but rather chosen dynamically to maximize the weighted loss by allocating the maximum weight(s) to the task(s) on which the model performs worst.

The main advantage of minimizing $\mathcal{L}_{\mathrm{DRO}}$ is that it ensures a certain level of performance on all tasks. However, it suffers from a number of drawbacks. First, it is sensitive to task difficulty: if one task $\ell_{\mathrm{hard}}$ is much more difficult than the others (i.e., $\min_{\boldsymbol{\theta}} \ell_{\mathrm{hard}}(\boldsymbol{\theta}) > \min_{\boldsymbol{\theta}} \ell_j(\boldsymbol{\theta})$ for all other tasks $j$), then $\mathcal{L}_{\mathrm{DRO}}$ is likely to get "stuck" trying to minimize $\ell_{\mathrm{hard}}$ at the detriment of other tasks.

In addition, DRO is agnostic to positive (or negative) interaction between tasks. In some cases, training on one task $\ell_i$ may have a positive effect on another task $\ell_j$ because they share some similarities—for example if they represent related languages. In such cases it may be desirable to train on both tasks rather than on one or the other (as it may reduce the variance of the gradient estimator), even if one has a higher loss than the other. On the other hand, when two tasks are at odds with each other (i.e., training on one task decreases performance on the other), the $\max$ objective in $\mathcal{L}_{\mathrm{DRO}}$ can lead to oscillatory behavior where the weights $\mathbf{w}$ alternate between two competing tasks.

## 2.4 LOOKAHEAD DRO

The core limitation of $\mathcal{L}_{\mathrm{DRO}}$ underpinning the aforementioned issues is the fact that it does not take the optimization procedure into account (i.e., it will bluntly try to minimize the worst loss). This is suboptimal because in certain cases, taking gradient steps on the worst task might hurt other tasks disproportionately. If the worst-performing task has converged already, this weight allocation will also miss out on optimizing other tasks, which can attain lower losses and are still not converged.

Formally, for a given value of the model parameters $\boldsymbol{\theta}_0$, $\mathcal{L}_{\mathrm{DRO}}$ chooses weights maximizing the model's loss $\mathbf{w}_0^* := \arg\max_{\mathbf{w}} \sum_i w_i \ell_i(\boldsymbol{\theta}_0)$. After training with weights $\mathbf{w}_0^*$, we obtain parameters $\boldsymbol{\theta}_1$ with loss $\mathcal{L}_{\mathrm{DRO}}(\boldsymbol{\theta}_1) = \max_{\mathbf{w}} \sum_i w_i \ell_i(\boldsymbol{\theta}_1)$. However, note that for $\boldsymbol{\theta}_1$, the worst-case weight allocation $\mathbf{w}_1^* := \arg\max_{\mathbf{w}} \sum_i w_i \ell_i(\boldsymbol{\theta}_1)$ might be different than $\mathbf{w}_0^*$, so we have no guarantees that $\mathcal{L}_{\mathrm{DRO}}(\boldsymbol{\theta}_1) \leq \mathcal{L}_{\mathrm{DRO}}(\boldsymbol{\theta}_0)$. In other words, training with the worst-case weights $\mathbf{w}_0^*$ does not guarantee that $\mathcal{L}_{\mathrm{DRO}}$ will decrease.

This observation suggests an alternative strategy: suppose $\mathbf{w}, \boldsymbol{\theta}_0 \to \boldsymbol{\theta}_{\mathbf{w},\boldsymbol{\theta}_0}$ is a fixed—and, for simplicity, deterministic—gradient descent algorithm minimizing the $\mathbf{w}$-weighted loss $\sum_i w_i \ell_i$ starting from parameters $\boldsymbol{\theta}_0$. Rather than choosing $\mathbf{w}$ to be the worst-case weights corresponding to $\boldsymbol{\theta}_0$, we should pick weights that minimize the worst-case loss of $\boldsymbol{\theta}_{\mathbf{w},\boldsymbol{\theta}_0}^*$, *i.e. after training*. This can be formalized as solving the min-max problem:

$$\min_{\mathbf{w} \in \Delta_N} \max_{\mathbf{v} \in \Delta_N} \sum_i v_i \ell_i(\boldsymbol{\theta}_{\mathbf{w},\boldsymbol{\theta}_0}) \tag{1}$$

Intuitively, decoupling the weights $\mathbf{w}$ used for training from the weights $\mathbf{v}$ used for computing $\mathcal{L}_{\mathrm{DRO}}(\boldsymbol{\theta}_{\mathbf{w},\boldsymbol{\theta}_0}^*)$ removes the restriction of just training on the worst-performing tasks and enables us to find a task weighting which is able to continuously minimize the worst-case loss. Note that in this case there might be multiple solution $\mathbf{w}$: for instance, it might be that one or more tasks have the same effect on the resulting worst-case task, and as such any permutation of their weights is an optimal solution. In such scenarios, we do not want to miss out on training on tasks on which performance

can still be improved (as long as worst-case performance decreases). This can be formulated as a maximum entropy principle (Jaynes, 1957): *if multiple optimal training weights exist, we should pick the one with the highest entropy.*

Because finding the optimal $\mathbf{w}$ necessitates "looking ahead" in the optimization trajectory from $\boldsymbol{\theta}_0$ to identify the DRO loss $\max_{\mathbf{v}} \sum_i v_i \ell_i(\boldsymbol{\theta}_{\mathbf{w},\boldsymbol{\theta}_0})$, we call this weight selection principle **Lookahead-DRO** (L-DRO). We provide a graphical illustration in Figure 1.

## 2.5 ONE-STEP LOOKAHEAD DRO

In practice we cannot explicitly solve the min-max game in Eq. 1 because estimating the payoff $\max_{\mathbf{v}} \sum_i v_i \ell_i(\boldsymbol{\theta}_{\mathbf{w},\boldsymbol{\theta}_0})$ would require unrolling the full optimization process to compute $\boldsymbol{\theta}^*_{\mathbf{w},\boldsymbol{\theta}_0}$. Instead, we simplify and take only one gradient step with a small learning rate $\lambda$, *i.e.* replacing $\boldsymbol{\theta}^*_{\mathbf{w},\boldsymbol{\theta}_0}$ with $\boldsymbol{\theta}_0 - \lambda \sum_i w_i \nabla \ell_i(\boldsymbol{\theta}_0)$. Such one-step approximations are often used in the meta-learning literature (Finn et al., 2017; Liu et al., 2018). Using the first order Taylor extension of $\ell_i$ enables us to rewrite the individual lookahead task losses as:

$$\ell_i(\boldsymbol{\theta}_0 - \lambda[\sum_{j=1}^{N} w_j \nabla \ell_j(\boldsymbol{\theta}_0)]) \approx \ell_i(\boldsymbol{\theta}_0) - \lambda \sum_{j=1}^{N} w_j \nabla \ell_i(\boldsymbol{\theta}_0)^{\mathsf{T}} \nabla \ell_j(\boldsymbol{\theta}_0)$$

The resulting payoffs are now linear in $\mathbf{w}$, which reduces the lookahead optimization problem in Eq. 1 into a bilinear matrix game:

$$\min_{\mathbf{w}} \max_{\mathbf{v}} \mathbf{v}^{\mathsf{T}} \mathbf{L} \mathbf{w} \qquad (2)$$

with a payoff matrix $\mathbf{L}$, called the "lookahead matrix", which can be decomposed as follows:

$$L_{ij} = \underbrace{\ell_i(\boldsymbol{\theta}_0)}_{\text{Loss of task } i} - \lambda \underbrace{\nabla \ell_i(\boldsymbol{\theta}_0)^{\mathsf{T}} \nabla \ell_j(\boldsymbol{\theta}_0)}_{\substack{\text{"interaction"} \\ \text{between tasks } i \text{ and } j}}$$

We can find a solution (Nash equilibrium) of this game using many available solvers such as fictitious play (Brown, 1951) or linear programming (Lemke, 1965). In such cases where multiple equilibria exist, we choose the one with the highest entropy as dictated by our maximum entropy principle. The lookahead learning rate $\lambda$ is treated as a hyperparameter. This results in a training algorithm which we outline in Algorithm 1.

---

**Algorithm 1:** Lookahead DRO

---

**Input:** Lookahead learning rate $\lambda$, initial parameters $\boldsymbol{\theta}_0$
$\boldsymbol{\theta} \leftarrow \boldsymbol{\theta}_0$
**while** *training is not over* **do**
  $\quad L \leftarrow [\ell_i(\boldsymbol{\theta}) - \lambda \nabla \ell_i(\boldsymbol{\theta})^{\mathsf{T}} \nabla \ell_j(\boldsymbol{\theta})]_{ij}$      `// Compute lookahead matrix`
  $\quad \mathbf{v}^*, \mathbf{w}^* \leftarrow \texttt{maxent\_nash\_solver}(L)$      `// Find Nash equilibrium of L`
  $\quad \boldsymbol{\theta} \leftarrow \texttt{optimizer\_step}(\sum_i w_i^* \ell_i)$    `// Optimizer step on `**`w`**`*-weighted loss`
**end**

---

To build more intuitions, we consider several edge cases and discuss how L-DRO handles them:

**Independent tasks.** We have $\nabla \ell_i^{\mathsf{T}} \nabla \ell_j = \mathbb{1}_{i=j}$. In this case, for small enough values of the lookahead learning rate, the optimal strategy for the row player ($\mathbf{v}$) is to pick the task $i^* = \arg\max_i \ell_i$ with the highest loss, and the only possible best response for the column player ($\mathbf{w}$) is to match this selection (since $L_{i^*i} = \ell_i^* - \lambda < L_{ij} = \ell_i$ for all $j \neq i$). In other words, we recover the original DRO objective of training against the worst-case loss. This can be interpreted as an "implicit assumption" of DRO that training on one task does not affect the others.

**Redundant tasks.** We have $\nabla \ell_i^{\mathsf{T}} \nabla \ell_j = 1$. In this case, the gradients of the tasks are perfectly aligned. In practice, this means that the payoff is independent of the value of $\mathbf{w}$, so any choice of the training weights is valid. By the maximum entropy principle, we should pick uniform weights, thus recovering the average task loss baseline.

**The highest loss task has already converged.** Consider the case where $\ell_0$ is maximal, but the model has already converged on task 0 (in other words $\nabla \ell_0 = 0$). In this scenario—assuming there

is no strong negative interaction among the remaining tasks—the highest loss will always be $\ell_0$ irrespective of the training weights $\mathbf{w}$. Again, by the maximum entropy principle, L-DRO advocates for picking uniform weights. This makes it possible to continue training on the remaining tasks even though $\ell_0$ has converged, bypassing the core limitations of distributionally robust optimization (DRO) identified in §2.3.

## 3 LOOKAHEAD DRO IN PRACTICE

### 3.1 GRADIENT AND LOSS NOISES

In most realistic machine learning scenarios we do not have access to the full losses $\ell_i(\boldsymbol{\theta})$ and their gradients. Rather, we compute stochastic estimates on minibatches $\hat{\ell}_i(\boldsymbol{\theta})$. In preliminary experiments, we find that this often leads to instability in the training algorithm, since even little noise in the lookahead matrix can lead to dramatically different Nash equilibria, which causes high fluctuations of the training weights $\mathbf{w}^*$.

To mitigate this issue, we adopt an online weight update procedure inspired by the online-DRO algorithm proposed by Sagawa et al. (2020). Instead of computing the optimal weights $\mathbf{w}^*$ at each step based on noisy losses and gradients, we keep a running set of weights $\mathbf{w}$, which we update jointly with model parameters $\boldsymbol{\theta}$. Specifically, we interleave gradient updates on $\boldsymbol{\theta}$ with exponentiated gradient steps (Kivinen & Warmuth, 1997) on $\mathbf{w}$ to minimize $\max_{\mathbf{v}} \mathbf{v}^{\mathsf{T}} L \mathbf{w}$. We show the weight update rule in Algorithm 2.

---

**Algorithm 2:** Lookahead DRO online weight update

**Function** `update_weights` (*Lookahead matrix* $\mathbf{L}$*, weights* $\mathbf{w}$*, weight update rate* $\eta$) :

$\quad \mathbf{v}^* \leftarrow \arg\max_{\mathbf{v}} \mathbf{v}^{\mathsf{T}} \mathbf{L} \mathbf{w}$       // Compute best response to $\mathbf{w}$

$\quad \tilde{w}_i \leftarrow w_i e^{-\eta[\mathbf{v}^{*\mathsf{T}} \mathbf{L}]_i}$       // Update weights

$\quad w_i \leftarrow \frac{\tilde{w}_i}{\sum_j \tilde{w}_j}$       // Re-normalize weights

$\quad$ **return** $\mathbf{w}$

---

In practice, we find that this online weight update alleviates the instability issues mentioned above. Note that the choice of exponentiated gradient is not arbitrary. It corresponds to a mirror descent step (Nemirovskij & Yudin, 1983) using the Kullback-Leibler divergence (Kullback & Leibler, 1951) as a distance function. Choosing weight updates that locally minimize the change in relative entropy plays into our maximum entropy guiding principle (Jumarie, 1990; Kapur & Kesavan, 1992).

### 3.2 LOOKAHEAD MATRIX COMPUTATION

The main computational bottleneck of L-DRO lies in the computation of the lookahead matrix $\mathbf{L}$. Recall that $L_{ij}$ can be decomposed in two terms: the loss $\ell_i$ and an "interaction" term $\nabla \ell_i(\boldsymbol{\theta})^{\mathsf{T}} \nabla \ell_j(\boldsymbol{\theta})$. While the former can be estimated essentially "for free" during the forward pass of back-propagation, computing the pairwise task gradients dot products is much more computationally demanding. Indeed, it necessitates computing (and holding in memory) $n$ model gradients, which can be prohibitive for larger models. To reduce the computational overhead, we only update this second term (which we call the "interaction matrix" $\mathbf{A} = [\nabla \ell_i(\boldsymbol{\theta})^{\mathsf{T}} \nabla \ell_j(\boldsymbol{\theta})]_{i,j=1...N}$) periodically, typically every 100 or 1000 training steps. In preliminary experiments, we also find that Lookahead-DRO works more reliably when the lookahead gradients of each task are re-normalized to one. This means that the interaction matrix is actually computed to be $A_{ij} = \nabla \ell_i(\boldsymbol{\theta})^{\mathsf{T}} (\nabla \ell_j(\boldsymbol{\theta})/\|\nabla \ell_j(\boldsymbol{\theta})\|)$. The effect of these two design choices are investigated in ablation studies in Appendices A.1 and A.2. We show the final algorithm for online L-DRO in Algorithm 3.

## 4 SYNTHETIC EXPERIMENTS AS A PROOF OF CONCEPT

In this section, we demonstrate the efficacy of L-DRO on a synthetic multitask setting. We design a toy experiment such that neither DRO nor static mixing is able to reach an optimal solution (Figure 2). We identify each task $i$ with a point $y_i$ on the two dimensional plane $\mathbb{R}^2$ and define their respective loss functions as the squared Euclidean distance of the model prediction $f(\boldsymbol{\theta})$ to $y_i$, $\ell_i(\boldsymbol{\theta}) = \|y_i - f(\boldsymbol{\theta})\|_2^2$. We distribute the tasks such that most points are clustered around $(-1, 0)$, to simulate a grouping of similar tasks (*e.g.* Indo-Europoean languages in our multilingual example). We then place a single point $y_{\text{outlier}}$ at $(1, 0)$ to emulate the presence of an outlier task.

---

**Algorithm 3:** Online lookahead DRO training loop

---

**Input:** Lookahead learning rate $\lambda$, weight update rate $\eta$, interaction matrix update interval $k$

$A \leftarrow 0$                                       `// Initialize interaction matrix at 0`

$\mathbf{w} \leftarrow \left( \frac{1}{N} \cdots \frac{1}{N} \right)$                        `// Initialize `**`w`**` as uniform weights`

**while** *training is not over* **do**

    **if** *i% k == 0* **then**

        $A \leftarrow \left[ \nabla \ell_i(\boldsymbol{\theta})^\intercal \frac{\nabla \ell_j(\boldsymbol{\theta})}{\|\nabla \ell_j(\boldsymbol{\theta})\|} \right]_{1 \leq i,j \leq n}$                 `// Update interaction matrix`

    $L \leftarrow \ell(\boldsymbol{\theta}) - \lambda A$                           `// Compute lookahead matrix`

    $\mathbf{w} \leftarrow \texttt{update\_weights}(\mathbf{w}, L)$       `// Update weights using Algorithm 2`

    $\boldsymbol{\theta} \leftarrow \texttt{optimizer\_step}(\sum_i w_i \ell_i)$     `// Optimizer step on `**`w`**`-weighted loss`

---

We simulate the increased difficulty of the outlier task by adopting a parameterization $f(\boldsymbol{\theta})$ that prevents the model from reaching any position closer than a fixed value $R$ from $y_{\text{outlier}}$. Specifically we adopt a polar parameterization $\boldsymbol{\theta} := (r, \phi)$ and set

$$ f(\boldsymbol{\theta}) = y_{\text{outlier}} + (R + r^2) \begin{bmatrix} \cos \phi \\ \sin \phi \end{bmatrix}. $$

This essentially creates a "forbidden region" of radius $R$ around $y_{\text{outlier}}$ of the space that the model cannot penetrate, thus lower-bounding $\ell_{\text{outlier}}$ to $R$. In experiments, we set $R = 1.25$.

Figure 2a depicts the optimization trajectories when the model is trained either with the average objective or with DRO. As expected, minimizing the average loss leads to a solution that performs well on the majority of tasks but poorly on the outlier. On the other hand, the DRO trajectory moves towards the latter but ends up stuck when it reaches a point where $\ell_{\text{outlier}}$ cannot be improved despite being the highest loss among all tasks.

On the other hand, while the L-DRO trajectory shown in Figure 2b follows a similar trajectory as DRO at first, it is able to reach a solution that still achieves the best possible $\ell_{\text{outlier}}$ while achieving lower loss on the other tasks. Also notice that the online variant follows a similar trajectory, despite the fact that it does not compute a Nash equilibrium at every step.

Finally, Figure 2c highlights the importance of the online variant in the presence of noise. We emulate the stochasticity of SGD by adding a small amount of Gaussian noise to both losses and gradients. Under these conditions, lookahead DRO becomes very unstable, an issue that is partially mitigated by the online update.

## 5 EXPERIMENTS

### 5.1 CIFAR100

As a first step towards a more realistic setting, we experiment with a multitask version of the CIFAR-100 dataset (Krizhevsky et al., 2009), in which the 20 coarse labels are treated as separate 5-way classification tasks (Rosenbaum et al., 2018; Yu et al., 2020). In this setting, we train a small CNN model with four convolution layers with 32 $3 \times 3$ filters each followed by a ReLU activation and a $2 \times 2$ max-pooling layer. The resulting representation is averaged and fed through a 3-layer fully connected feed-forward network with hidden dimension 128 and ReLU activations. Predictions are made using a shared softmax layer across all tasks (*i.e.* across all 100 classes).

In addition to L-DRO, we report results for three baselines. First **static-mixing**, where all tasks are weighted equally (§2.2). Another natural baseline is **DRO** (§2.3) using the online algorithm proposed by Sagawa et al. (2020). Finally we compare to **PCGrad** (Yu et al., 2020), a multitask learning approach, which uses gradient level projections to mitigate interferences between tasks during learning. We train using regular stochastic gradient descent with a batch size of 128, a learning rate of 0.025 and with Nesterov accelerated gradients with a momentum rate of 0.9. For DRO, we sweep over 5 values of the hyper-parameter which controls the online weight update rate $\{0, 0.0001, 0.001, 0.01, 0.1\}$. For L-DRO, we sweep over 3 values of the weight update rate $\{0.01, 0.001, 0.0001\}$ and 3 values of the lookahead learning rate $\{0.001, 0.01, 0.1\}$. For Static mixing and PCGrad, there are no additional parameters to sweep over.

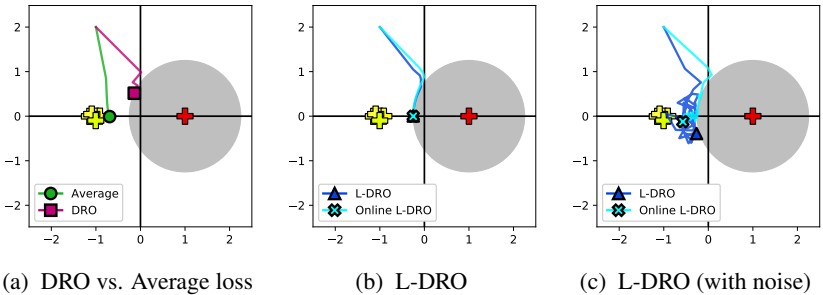

(a) DRO vs. Average loss     (b) L-DRO     (c) L-DRO (with noise)

Figure 2: Synthetic multitask experiment. The outlier task is shown in red while the others are in yellow. The grayed out area represents the region in space that cannot be attained by the model with our chosen parameterization

To gain a better understanding of the various trade-offs of each approach in terms of average and worst accuracy, we do not report a single number. Rather, we periodically evaluate the trained models on the test set and report the Pareto frontier of all checkpoints (across epochs and hyper-parameter configurations) of each method in the two dimensional plane of worst and average accuracy.

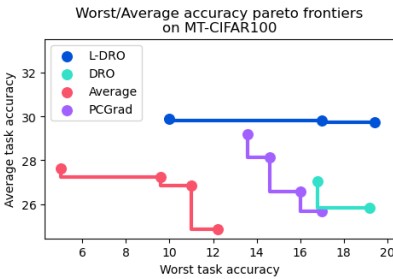

We show the results in Figure 2. We can see that L-DRO (in blue) consistently achieves a good trade-off between the two metrics, as its Pareto frontier is situated in the upper right corner of the space (high average and worst-case accuracy). In particular it is able to reach similar worst-case performance as DRO, without incurring a hit

Figure 3: Average/worst task accuracy Pareto frontiers on multitask CIFAR-100

in terms of average performance. We find that this trend is consistent across different random seeds (see Appendix B.1).

An important difference in our experimental setting with Yu et al. (2020) is that the model we train is comparatively much smaller. In fact, as we increase the size of the model, we observe that the difference between individual method grows thinner (detailed results are depicted in Appendix B.3). This finding suggests that L-DRO is primarily useful when the model's capacity is a limiting factor compared to the number of tasks and the difficulty of individual tasks, which is not necessarily the case for the simple CIFAR-100 benchmark.

## 5.2 MULTILINGUAL LANGUAGE MODELING

To evaluate L-DRO in a real-world scenario, we turn to large-scale multilingual language modeling. As NLP models are increasingly applied to new settings, pretraining on many languages is becoming a common scenario (Conneau et al., 2020; Xue et al., 2021). Improved pretraining performance has generally been found to correlate with better downstream performance (Raffel et al., 2020).

Furthermore, multilingual language modeling is a particulary interesting testbed because it exhibits various properties of real world multitask learning scenarios. For example, there is a large data imbalance across tasks: the highest resource language (English) contains $55\times$ more data than the lowest resource language (Yorùbá) in our training corpus. Additionally, there is a difference in difficulty between the various languages. Formally, for a parametric language model $p_{\boldsymbol{\theta}}$, its loss on the $i$-th language is the cross-entropy $\mathbb{H}(p_i, p_{\boldsymbol{\theta}})$ with the underlying distribution of the language, $p_i$. This can be re-written as $\mathbb{H}(p_i) + D_{\mathrm{KL}}(p_i||p_{\boldsymbol{\theta}})$. Since the KL divergence is non-negative, the loss of each language is lower-bounded by the language-specific entropy term $\mathbb{H}(p_i)$, irrespective of the capacity or the fitness of the model. In other words, languages with higher entropy will be more difficult to learn (Mielke et al., 2019).

### 5.2.1 EXPERIMENTAL SETTING

We train a large autoregressive transformer model (Vaswani et al., 2017) on the multilingual C4 dataset (mC4; Xue et al., 2021). We limit ourselves to the 40 languages that are present in the

XTREME benchmark (Hu et al., 2020), which belong to 12 different language families and are written in 12 different scripts. Out of all languages, 40% are Indo-European and 42.5% are written in the Latin script.[1] The mC4 dataset comes with a canonical "training" and "evaluation" split. In order to distinguish between validation and test set, we modify the evaluation split as follows: for each language we split the evaluation set into a validation and test set containing up to 5M tokens each (using mT5 subword tokenization; Kudo 2018), or half the entire evaluation set, whichever is largest. We limit the size of the validation and test sets to facilitate running validation of our models.

We train each model for $300,000$ steps with a batch size of 256 elements of 512 tokens using the Adam optimizer (Kingma & Ba, 2014) and a cosine annealed learning rate schedule (Vaswani et al., 2017). For each run we select the best model in terms of average validation loss on all languages. For evaluation, we set the sequence length to 1024.

In this experiment we compare against three baselines:[2]

- **Static mixing** (§2.2). We try 3 different values for the inverse temperature hyper-parameter $\alpha$: 0 (uniform sampling), 0.3 (a common value in the literature; Xue et al. 2021) and 1 (purely proportional sampling)
- **DRO** (§2.3), which minimizes the worst-case loss among all languages using the online algorithm from Sagawa et al. (2020). We sweep over the weight learning rate $\eta \in \{0.01, 0.001, 0.0001\}$.
- **Baselined-DRO**, a variation on DRO proposed by Oren et al. (2019) where the losses of each language $i$ is adjusted with a "baseline" loss to account for the entropy term which biases DRO towards selecting languages that are inherently more difficult. In practice, we estimate $\mathbb{H}(p_i)$ by training a separate *monolingual* model on each language, $p_{\boldsymbol{\theta}_i^*}$ and using its cross-entropy $\mathbb{H}(p_i, p_{\boldsymbol{\theta}_i^*}) = \min_{\boldsymbol{\theta}} \mathbb{H}(p_i, p_{\boldsymbol{\theta}})$ as a proxy. We then train to minimize $\max_i \ell_i(\boldsymbol{\theta}) - \mathbb{H}(p_i, p_{\boldsymbol{\theta}_i^*})$ using the same algorithm as DRO, sweeping over the same values of the weight learning rate. Note that this particular variant necessitates training an additional model for each language.

### 5.2.2 RESULTS

We report results in Table 1 in terms of perplexity (Jelinek et al., 1977), the standard metric for language modeling. Perplexity is the exponential of the (token-wise) cross-entropy. As a result, perplexities of different languages are not directly comparable. Languages with inherently higher entropy will tend to contribute more to the average (or worst) perplexity. Therefore, we also report average and worst-case perplexity *relative* to a monolingual model. Specifically, the relative perplexity of a model $\boldsymbol{\theta}$ on language $i$ is defined as:

$$\text{Relative perplexity}(\boldsymbol{\theta}, i) = \frac{\text{Perplexity}(\boldsymbol{\theta}, i) - \text{Perplexity}(\boldsymbol{\theta}_i^*, i)}{\text{Perplexity}(\boldsymbol{\theta}_i^*, i)} \quad (3)$$

where $\boldsymbol{\theta}_i^*$ is a monolingual model trained on language $i$. In other words, relative perplexity of a given model on a language quantifies how much worse the multilingual model is compared to a monolingual model trained on the same language (with the same data).

Our results show that L-DRO reaches the best average score both in terms of perplexity and relative perplexity. While DRO is by far the best alternative in terms of worst perplexity, it ranks lowest in terms of relative perplexity. This is because its focus on languages with high perplexity comes at the detriment of low-entropy languages. Unsurprisingly, proportional mixing based methods ($\alpha = 0.3$ or 1) perform very well on high resource languages (*e.g.* English or French) but poorly on almost all other languages, leading to low performance along almost all metrics.

On the other hand, L-DRO performs much better in terms of relative perplexity: it outperforms both static mixing and DRO in terms of average and worst-case performance. Note that Baselined-DRO, which takes the entropy of individual languages into account when computing the maximum, reaches the lowest worst-case relative perplexity. However, it is a much more computationally demanding approach: it requires training 39 additional monolingual models to estimate the language entropies. In comparison, L-DRO comes in as a close second in terms of worst-case relative perplexity, with a comparatively much smaller computational overhead (periodic computation of $\mathbf{L}$), which makes it

---

[1] In experiments, we omit the Tagalog language, which is not included in mC4.

[2] We do not compare against PCGrad due to its underwhelming performance in the CIFAR-100 experiments and the prohibitive cost of its gradient projection step in this setting.

|  | absolute perplexity | | relative perplexity | |
|---|---|---|---|---|
|  | average | worst | average | worst |
| Static Mixing ($\alpha = 1$) | 48.7 | 135.5 | 358.3 | 1170.7 |
| Static Mixing ($\alpha = 0.3$) | 24.0 | 41.2 | 130.6 | 213.7 |
| Static Mixing ($\alpha = 0$) | 23.9 | 48.7 | 128.2 | 198.1 |
| DRO | 24.5 | **35.1** | 142.7 | 286.7 |
| Baselined-DRO | 23.9 | 50.7 | 127.3 | **168.5** |
| L-DRO | **23.6** | 55.5 | **126.0** | 191.7 |
| Monolingual | 10.6 | 25.3 | 0 | 0 |

Table 1: Results on the multilingual language modeling experiment. We report results both in terms of absolute and relative perplexity.

a much more appealing alternative. A more detailed discussion of the computational overhead of L-DRO can be found in appendix C.

## 6    RELATED WORK

Much work in multi-task learning has focused on improving the optimization process when learning many tasks. Some prior work (Sener & Koltun, 2018; Lin et al., 2019; Ma et al., 2020) casts multi-task learning as multi-objective optimization, with the goal of finding a Pareto-optimal solution. Other work proposes to directly modify the gradients. In order to minimize gradient interference, Yu et al. (2020) project conflicting gradients on the normal plane of the other. Wang et al. (2021) alter both the direction and magnitude of gradients to achieve a certain gradient similarity.

Many other methods have been proposed to weight the task losses according to different criteria such as uncertainty (Kendall et al., 2018), reward magnitude (Hessel et al., 2019), and learning speed (Chen et al., 2018; Liu et al., 2019; Zheng et al., 2019). Most of the latter methods increase a task's loss weight when the learning speed for that task is low. Similar to learning speed, a few approaches (Guo et al., 2018; Jean et al., 2019) weight task losses based on performance, focusing on optimizing tasks with poor performance. Among these, our proposed method is most similar in spirit to performance weighting-based methods. Such methods, however, do not consider the optimization procedure and are not aware of task similarity. Loss weighting can be seen as a continuous relaxation of the task scheduling and sampling so that most task scheduling methods can be easily adapted to loss weighting methods, and vice versa (Ruder, 2019; Crawshaw, 2020). Kiperwasser & Ballesteros (2018) proposed different predefined sampling schedules while Sanh et al. (2019) proposed a proportional sampling strategy. In multilingual language modeling and machine translation (Aharoni et al., 2019; Conneau et al., 2020; Xue et al., 2021), proportional sampling with a temperature is typically employed.

The distributional robustness literature focuses on training models that perform well on various domains (Ben-Tal et al., 2009; Rahimian & Mehrotra, 2019). This is generally performed by minimizing the worst-case expected risk over a pre-determined family called the "uncertainty" set. There has been substantial work on formulating appropriate uncertainty sets (Hu & Hong, 2013; Gao & Kleywegt, 2016; Levy et al., 2020). Most relevant to our multitask setting are group-structured uncertainty sets, where the maximimum is taken over a mixture of sub-populations (Oren et al., 2019; Sagawa et al., 2020; Zhou et al., 2021).

## 7    CONCLUSION

In this paper, we argued for the importance of looking at worst-case performance when training and evaluating multi-task models, especially in cases where the distribution of tasks is not homogeneous (e.g., when some tasks are more difficult than others) or when there is an imbalance towards certain types of tasks. We proposed L-DRO, which enables efficient optimization of the worst-case loss without the pitfalls of existing DRO-based max-loss minimization approaches. We demonstrated the benefits of L-DRO in a synthetic scenario as well as in realistic image classification and language modeling experiments. Overall, we encourage the community to evaluate not only on homogeneous task collections (e.g., CIFAR-100) but also "heterogeneous" multitask learning scenarios (e.g., multilingual language modeling) when assessing the performance of a multitask model.

## ETHICS STATEMENT

Measuring models only based on their average performance obfuscates their limitations on under-represented subpopulations and contributes to the marginalization of said groups. Developing algorithms that work well not just for the average user but for every type of user is thus important from an ethical perspective.

## REPRODUCIBILITY STATEMENT

We take several steps to ensure reproducibility of our work. Pseudo-code describing the final version of our algorithm is given in Algorithm 3. Moreover, we detail experimental settings and model hyper-parameters in §5. Finally, all our experiments are performed on datasets that are openly available: CIFAR-100[3] (Krizhevsky et al., 2009) and mC4[4] (Xue et al., 2021).

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

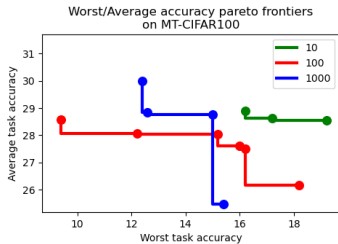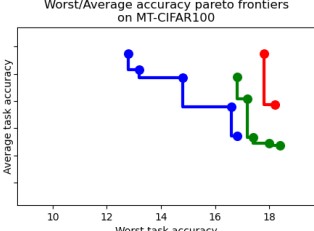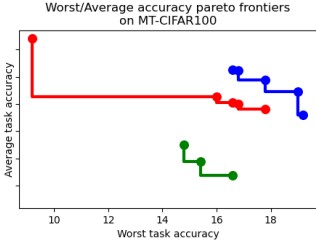

Figure 4: Average/worst task accuracy Pareto frontiers on multitask CIFAR-100 for L-DRO with different interaction matrix update intervals. Each plot corresponds to a different random seed.

# A    ABLATION STUDIES

## A.1    EFFECT OF PERIODIC MATRIX UPDATES

We investigate the effect of periodic matrix updates on the results of L-PDRO. Figure 4 reports the Pareto frontier of L-DRO over a range of hyper-parameters $\eta$ and $\lambda$ on the CIFAR-100 experiment from Section 5.1. We compare frontiers for 3 different interaction matrix update intervals: 10, 100 and 1000, and report results for 3 random seeds.

Although there is some variability across different runs, we do not observe that larger intervals are consistently worse than smaller intervals. That being said, in experiments with larger models, we have observed some instability with the larger update intervals. This is evidenced in Figure 5a for a CIFAR-100 run with the larger CNN model used in Section B.3.

## A.2    EFFECT OF RENORMALIZATION

Note that there of the two components in each element in the interaction matrix $A_{ij} = \nabla\ell_i^\intercal \nabla\ell_j$ and we only normalize one ($\nabla\ell_i^\intercal \nabla\ell_j \rightarrow \nabla\ell_i^\intercal \frac{\nabla\ell_j}{|\nabla\ell_j|}$).

This can be interpreted as the first order approximation of taking a lookahead step along the *normalized* task gradients:

$$\ell_i(\theta_0 - \lambda[\sum_{j=1}^{N} w_j \frac{\nabla\ell_j(\theta_0)}{\|\nabla\ell_j(\theta_0)\|}]) \approx \ell_i(\theta_0) - \lambda \sum_{j=1}^{N} w_j \nabla\ell_i(\theta_0)^\intercal \frac{\nabla\ell_j(\theta_0)}{\|\nabla\ell_j(\theta_0)\|}$$

Our intuition with this re-normalization trick is to help mitigate the instability caused by the "staleness" of the interaction matrix with periodic updates. Indeed, the approximation error of using a stale interaction matrix comes from two sources: outdated alignment between the gradient (cosine similarity) and outdated gradient norms. With the re-normalization trick, we mitigate the latter by factoring out the norm of the lookahead step while keeping the original interpretation (as explained above).

In practice, we do observe that this helps improve stability for larger matrix update interval. Indeed, in an ablation study in CIFAR-100 (see Figure 5b), we find that the use of the re-normalization trick removes the instability observed with larger update intervals (1000).

# B    ADDITIONAL CIFAR-100 RESULTS

## B.1    MULTIPLE RANDOM SEEDS

Figure 6 depicts results on the CIFAR-100 dataset for 3 different random seeds. We find that L-DRO consistently outperforms baselines.

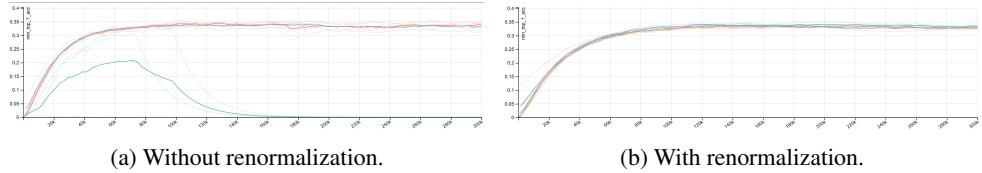

|                                |                               |
| ------------------------------ | ----------------------------- |
| (a) Without renormalization.   | (b) With renormalization.     |

Figure 5: Worst-case accuracy training curves for CIFAR-100 with $\eta = 1$, $\lambda = 0.001$ and interaction matrix update intervals 10 (blue), 100 (orange) and 1000 (green). The curves are smoothed with a running exponential average for readability.

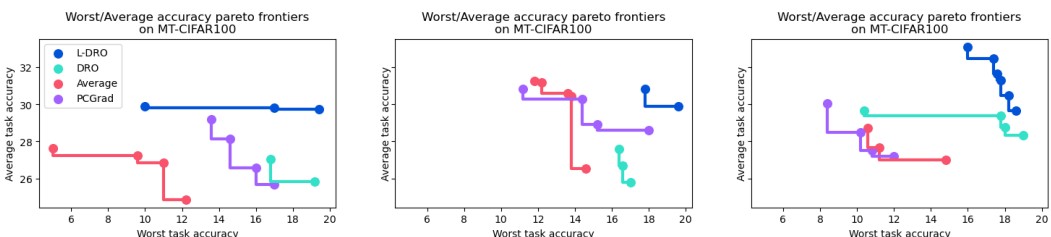

Figure 6: Average/worst task accuracy Pareto frontiers across 3 random seeds on multitask CIFAR-100

## B.2 Additional Baseline

Another natural baseline for trading off worst and average accuracy is to optimize a weighted average of average and worst-case task loss:

$$\min_{\boldsymbol{\theta}} \alpha \times \frac{1}{N} \sum_{i=1}^{N} \ell_i(\boldsymbol{\theta}) + (1 - \alpha) \times \max_{i} \ell_i(\boldsymbol{\theta})$$

with some interpolation parameter $\alpha$. A downside of this approach is that, for $\alpha > 0$, the model cannot, in general, reach the minimal worst-case loss because of the nonzero "average loss" component.

Figure 7 shows results including this additional baseline with $\alpha = 0.5$ (equal weights to both the average and worst-case loss). We observe that while interpolating between DRO and Static mixing is occasionally better than either on their own, it doesn't achieve the same average/worst-case performance trade-off as L-DRO.

## B.3 Experiments with larger CNN

In Figure 8, we report CIFAR-100 results across 3 random seeds using a larger CNN model: 3 layers of 160 $3 \times 3$ filters each followed by a ReLU activation and a $2 \times 2$ max-pooling layer, followed by a 2 layer multilayer perceptron with hidden dimension 320 and ReLU activations leading into a final fully connected layer mapping to the 100 classes. We find that the difference between the

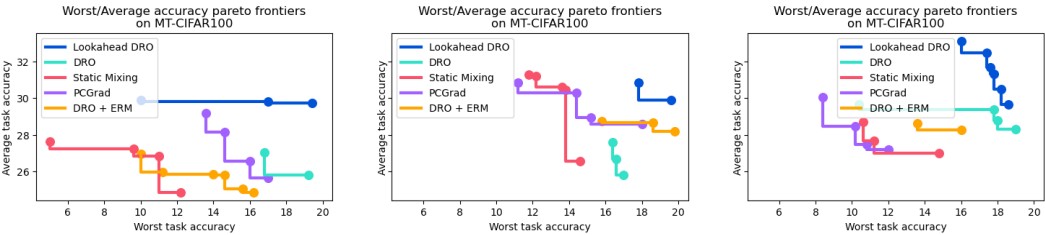

Figure 7: Average/worst task accuracy Pareto frontiers across 3 random seeds on multitask CIFAR-100, including additional baseline.

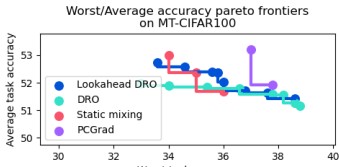 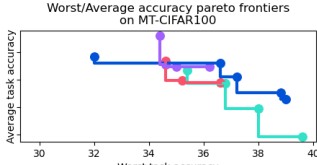 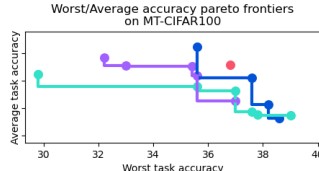

Figure 8: Average/worst task accuracy Pareto frontiers across 3 random seeds on multitask CIFAR-100

various method vanishes. Although L-DRO tends to achieve similar robust accuracy as DRO at higher average accuracy, the difference is small but not consistent across the different runs.

## C  TIME AND MEMORY OVERHEAD OF L-DRO

The main overhead of L-DRO is the computation of the interaction matrix $A$. The most straightforward approach for doing so is to compute the $N$ task gradients and take their pairwise dot-product. The overhead incurred is equivalent to taking $N$ forward/backward passes (one to obtain each of the task gradients), as the cost of the pairwise dot-product is marginal in comparison. In theory, this means that training with L-DRO becomes approximately $N/T$ times slower, where $T$ is the interaction matrix update interval.

We employed this strategy for the CIFAR-100 experiments in Section 5.1. With $N = 20$ and $T = 100$ in that setting, the theoretical slowdown is approximately $\times 1.2$. In practice, for our biggest CNN model on CIFAR-100, training speed was around 27 steps/s for L-DRO vs 28.5 steps/s for ERM (for reference, PCGrad ran at 21 steps/s). This means that the effective slowdown is approximately 1.06. We attribute this improvement over the theoretical limit to the optimized Jacobian computation in our JAX implementation.

The downside of this first approach is that it necessitates holding $N$ gradients in device memory. This can quickly become cumbersome when the number of tasks and the size of the model are large. For the language modeling experiment in Section 5.2, we adopt the alternate —more memory efficient— strategy of re-computing each dot-product separately. This means the memory overhead drops to only holding 2 gradients in memory at each time, but we need to compute $O(n^2)$ backward passes. In practice for the language modeling experiment, there $N = 49$ tasks and, and the interaction matrix was updated every $T = 1000$ steps, leading to a theoretical slowdown of approximately $\times 3.4$. By optimizing the interaction matrix computation (*e.g.* computing only the upper triangle, using a ($2\times$) smaller batch size to estimate the gradients), the final training speed was 2.2 steps/s for L-DRO, as opposed to approximately 3.4 steps/s for ERM, *i.e.* an effective slowdown of $\times 1.55$.

Note that another way to reduce device memory usage would be to use the first approach and offload the computation of the gradient dot-products to the CPU (where memory is much less of an issue), in which case the additional overhead comes from the cost of moving gradients from device to CPU as well as computing large dot-products on CPU. However, we did not experiment with this last approach.

