# OpenReview forum: "Balancing Average and Worst-case Accuracy in Multitask Learning"
_ICLR.cc/2022/Conference — ICLR 2022 Submitted_

### Official Review · Reviewer_j3ej · 2021-10-26

**Correctness:** 3
**Technical Novelty And Significance:** 3
**Empirical Novelty And Significance:** 2
**Recommendation:** 6
**Confidence:** 3

**Main Review:**

Strengths:

1. The idea sounds legit, interesting and novel.

2. The writing is clear and easy to follow.

Weaknesses:

1. The empirical improvement does not seem to be very significant, especially on large models.

2. Another concern is with all the approximations, it is no longer clear whether the merit of the proposed method is still held, i.e., can the proposed L-DRO guarantee that $L_{DRO}(\theta_1) < L_{DRO}(\theta_0)$, empirically?

Questions:

1. Does Figure 5 imply the advantage of the proposed L-DRO is mitigated by model with larger capacity?

2. How large is the variance of the experiments? Is 3 runs enough?

3. Can the authors visualize the weights in algorithm 2? Will the trick make the weight $w$ almost consistent across the training?

---- Post Rebuttal

I thank the authors for the response. I would like to keep my original rating as weakly accept, with the reason that 1) I think the novel viewpoint could potentially benefit the community and 2) the diminishing performance of the proposed method when the model gets larger makes the approach less interesting.

**Summary Of The Paper:**

This submission proposes Lookahead-DRO, of which the core idea is to do DRO on the updated weights, i.e., to choose a weight that minimizes the DRO loss in the next step. The computation is not tractable so the authors propose to use first order Taylor expansion approximation. Experiments on synthetic datasets and two real-world datasets demonstrate its effectiveness.

**Summary Of The Review:**

Interesting idea, good writing, moderate empirical results

---

> ### Author Response · Authors · 2021-11-22
> **Response to Reviewer j3ej**
>
> We thank the reviewer for their encouraging feedback. We respond to their various questions below:
>
> > The empirical improvement does not seem to be very significant, especially on large models.
>
> For CIFAR-100, we indeed find that all methods (not only L-DRO) perform roughly the same if the model is large enough. Please see our response to all reviewers for more discussion of this result.
>
> Regarding the language modeling experiments, while the numerical improvements are small, they are still meaningful with regards to the amount of data being evaluated on ($\approx 5M$ tokens per language).
>
> > Another concern is with all the approximations, it is no longer clear whether the merit of the proposed method is still held, i.e., can the proposed L-DRO guarantee the DRO loss decreases
>
> Empirically, we do observe that the DRO loss tends to decrease during training using L-DRO. This can be seen in the toy experiments in Figure 2 (b). In more realistic experiments, we do find that there is some noise in the trajectory of all losses, due to (1) the Taylor expansion approximation and (2) stochasticity in the gradient estimates.
>
> > Does Figure 5 imply the advantage of the proposed L-DRO is mitigated by model with larger capacity?
>
> Indeed. Again, we refer to our response to all reviewers for further discussion of this result
>
> > How large is the variance of the experiments? Is 3 runs enough?
>
> We find that training curves are relatively stable across the 3 reruns, at least on CIFAR-100 (due to the large size of mC4, running multiple LM experiments is difficult).

---

### Official Review · Reviewer_Bw6i · 2021-11-02

**Correctness:** 4
**Technical Novelty And Significance:** 3
**Empirical Novelty And Significance:** 2
**Recommendation:** 5
**Confidence:** 4

**Main Review:**

Overall, the idea of this paper is well presented and intuition is well explained. Below please find my detailed comments.

Major:
1. Seems for most cases when the worst task’s loss is larger than other losses with a certain margin. This algorithm only performance gradient descent on this worst task.

The advantage of this algorithm only appears when the top-n worst tasks are very close to each other. In this case, considering task interactions might help. But I was wondering whether such a scenario happens often in practice?

2. Seems there is a significantly simpler algorithm functioning similar to the proposed one:
At each iteration, we find the worst task, performance gradient on this worst task. If the gradient is not zero, we apply the gradient. Else, we apply the uniformly weighted gradient of all the other tasks.

There might be a slight difference between this very simple algorithm and the proposed one. I was wondering how do they compare.

3. No convergence guarantee is shown here. I.e., does this algorithm converges to the Pareto stationary set? What’s the convergence rate interns of worst-case loss?

Minor:
1. There are various criteria to decide the final objective for multi-task learning. There is not much explanation why minimizing the worst loss is a good choice.


**Summary Of The Paper:**

This paper propose one-step look ahead DRO for multitask learning considering the worst task’s performance as the objective. Due to the task’s interaction with each other, directly using gradient-based approach is suboptimal. To overcome, look-ahead DRO estimates the task interaction using Taylor expansion and returns the locally optimal weighting by solving a linear min-max subproblem. Several tricks for mini-batch and online updating version of DRO are proposed.

**Summary Of The Review:**

I think the general idea of this paper is interesting. However, there are still several important points that need to be clarified. I am happy to have more discussion and adjust my score accordingly if things are properly addrssed.

---

> ### Author Response · Authors · 2021-11-22
> **Response to Reviewer Bw6i**
>
> We thank the reviewer for their helpful feedback. We respond to their core concerns below:
>
> > For most cases when the worst task’s loss is larger than other losses with a certain margin, this algorithm only performs gradient descent on this worst task.
>
> This is not necessarily the case: refer for instance to the third example at the end of Section 2.5: if the gradient of the highest loss is 0, then L-DRO will train on all tasks jointly irrespective of the the value of the highest loss.
>
> This is showcased in practice in our synthetic experiment in Section 4: in figure 2.b, L-DRO ends up training on the yellow tasks whenever it reaches the boundary of the gray region, even when the loss of the red task is higher.
>
> > What about a simpler baseline where we train on all tasks when the gradient of the worst-case task is 0?
>
> We thank the reviewer for this interesting suggestion. Indeed, this simpler algorithm addresses one of the key issues of DRO we pointed out in the paper: that DRO can get stuck optimizing the worst-case loss.
>
> A key deficiency of this approach is that it requires the gradient of the worst-case loss to be exactly zero (or below some threshold), which is rarely the case in stochastic gradient descent when all the gradients we get are minibatch-level estimates.
>
> In preliminary experiments on our toy setting, we observe that this baseline indeed reaches the same pareto optimal solution as L-DRO *in the absence of noise*. However, whenever gradient noise is introduced, the algorithm tends to get “stuck” optimizing the worst-case loss. This does not happen with L-DRO which can still choose to perform steps on other tasks even when the worst performing task’s gradient isn’t 0.
>
> Possibly, these failings could be remedied by tuning the gradient norm threshold, computing running averages, etc… but this takes the method beyond the “simple baseline” and outside the scope of our paper.
>
> > No convergence guarantee is shown here. I.e., does this algorithm converges to the Pareto stationary set? What’s the convergence rate interns of worst-case loss
>
> We agree with the reviewer that our contribution is mostly empirical. Indeed, there is little we can say in the case where the task losses are not convex, which is the more interesting application of L-DRO. Despite this fact, we hope that our case study of failure cases of DRO in Section 2 and our positive experimental results are sufficient to motivate the use of L-DRO in practical scenarios.
>
> > There are various criteria to decide the final objective for multi-task learning. There is not much explanation why minimizing the worst loss is a good choice.
>
> We agree with the reviewer that worst-case performance is but one of the many criteria that can be used to evaluate multitask models. As alluded to in the introduction, there are specific instances where worst-case performance _is_ an important objective, eg. when the distribution of tasks is biased towards certain types of tasks or data, at the detriment of others (for example our multilingual NLP example).
>
> While we did cite some related work in multitask learning which deals with different objectives (eg. Sener & Koltun, 2018 and follow ups), we are happy to include additional pointers that the reviewer feels are lacking.

---

### Official Review · Reviewer_TtaP · 2021-11-03

**Correctness:** 3
**Technical Novelty And Significance:** 2
**Empirical Novelty And Significance:** 3
**Recommendation:** 6
**Confidence:** 4

**Main Review:**

**Strengths:** The problem is very important. The authors bring the attention of the community of an important shortcoming of both average loss and worst case loss in the settings when there is imbalance in task distribution. The paper for most part is well written.

**Weaknesses:** I have several concerns that I highlight below:
1. **Look ahead DRO and role of deterministic algorithm** The exact approach of look ahead DRO is presented in Section 2.4. The approach has several problems that are not clear. If we assume that the deterministic algorithm finds $\theta_{w,\theta_0}$ achieves the global minimum for the weight $w$, then I cannot see how this approach is any different from doing regular min-max optimization. The reason I say is that if you do weighted sum risk minimization, then the loss values for different tasks achieved at the optima for different values of weights cover the entire Pareto boundary of the graph of loss of tasks. The problem amounts to selecting the point on the Pareto boundary that achieves the best worst case risk.
The authors should show using at least simple 2D illustrations, what point on the Pareto plane DRO achieves, and what point exact look ahead DRO intends to achieve, and contrast with standard sum risk minimization.
I suspect the exact lookahead DRO is implicitly assumed that it will not find the exact minimum. If that is the case, then there should be a very detailed discussion on it.

2. **Two baseline approaches** The title of the paper is about balancing worst case accuracy and average accuracy. The title itself suggests that the authors should compare with a simple combination of minimization of worst case together with average risk. This can be stated as
$$min_{\theta}\Big( max_{k}l^{k}(\theta) + \lambda \sum_{k} l^{k}(\theta)\Big)$$
This gradients of the above can done with the approach of either Oren et al. and also perhaps Sagawa et al.
The above objective exactly tries to optimize what the authors require. It is not clear why there are no comparisons with the above approach since it is the most natural baseline to study.

The objective in equation (8) in http://proceedings.mlr.press/v139/krueger21a/krueger21a.pdf can also solve the problem that the authors propose. The objective minimizes the average loss across tasks and regularizes it using the variance of the loss across tasks. Hence, if for each task if you compute the loss as the average loss over the data by dividing it by the size of the data for that task, this would be a useful approach to compare against.

3. **Regarding experiments** In the main real experiment, the baseline-DRO performs the best. The authors claim that its much more computationally expensive. I have two main points to make here
a) Baseline DRO is still a DRO based approach. Why is it able to achieve a much better tradeoff? The entire paper up until this section says that you will achieve a better tradeoff irrespective of the compute budget. This seems to be going against the main idea presented. Even your synthetic experiment at least agrees with your main hypothesis.

b) You claim that the computational expense is much higher. If that is the case and that is the main source of gain, why not report the wall clock computation times to make the point clear? There is one more reason I ask this is because in Oren et al. (Section 4.2) the authors state " In practice, we use a bigram model, which was fast enough to scale and worked sufficiently well in experiment". What you say seems to contradict their claim.








**Summary Of The Paper:**

In this work, the authors study the problem of multi-task learning in the presence of heterogenous tasks when there is an imbalance in the distribution of tasks. In such settings, we can learn models for multi-task learning using two approaches -- a) learn to minimize average loss across tasks, b) learn to minimize the worst case loss. Turns out both the approaches are not ideal. Minimizing average loss can compromise one task a lot and minimizing worst case loss can focus only on one very difficult task at the expense of improvements possible in other tasks. The authors propose a look ahead distributionally optimization based approach to balance the two -- worst case performance and the average performance. In the proposed approach, instead of solving a standard min-max optimization, the authors propose to constrain the min learner to find the optimal weight combination for weighted risk minimization, which leads to the minimum worst case risk.
The authors carry out synthetic and real datasets to demonstrate the efficacy of the approach.

**Summary Of The Review:**

Overall, the paper addresses an important problem. It proposes an interesting approach to solve it as well. The synthetic experiment is promising. However, there are quite a few weaknesses -- the target that exact lookahead DRO approach seems to be no different than regular DRO (as I explained above), missing natural baselines (without such baselines its hard to even understand if such a lookahead is needed or not),  problems with real data experiments. If the authors can effectively address these concerns, I would be willing to change my score.

---

> ### Author Response · Authors · 2021-11-22
> **Response to Reviewer TtaP**
>
> We thank the reviewer for their insightful comments
>
> > Look ahead DRO and role of deterministic algorithm
>
> We thank the reviewer for bringing up this point which highlights an imprecision in our formulation. Indeed, Lookahead-DRO makes sense specifically for gradient-based training. We rewrote Section 2.4 to emphasize this point:  $\theta^*$ is assumed to be the result of gradient descent on the $w$-weighted loss.
>
> With this clarification in mind, we can show that L-DRO is *not* equivalent to DRO.
>
> Indeed, consider the case where all losses are convex and subdifferentiable. In that scenario, $\theta^*_{w,\cdot}$ does converge to a global optimum of the $w$-weighted loss irrespective of the starting point $\theta_0$ (by convexity). In addition, any optimum of DRO can be reached by minimizing the $w$-weighted loss for a well-chosen value of $w$ (see Proposition 1 in Sagawa et al. (2019)).
>
> However, even in that scenario, DRO is not equivalent to L-DRO due to the maximum entropy criterion. Indeed, consider a case where there are more than one solution with best worst-case risk. To each of these solutions (\theta^*) corresponds one or more weightings $w^*$ such that $\theta^*=\text{argmin}\_{\theta}\sum_iw^*_i\ell_i(\theta)$. However, in standard min-max optimization, there is no criterion to distinguish among these.
>
> On the other hand, L-DRO will only pick the one of the DRO solutions that corresponds to a global minimizer of a $w^*$ weighted loss where $w^*$ has *maximal* entropy. In other words, L-DRO will choose $w^*$ that is closest to uniform weights (in the KL sense).
>
> This is actually shown empirically in our toy experiment in Section 4 where all losses are convex: there is a continuum of DRO solutions on the boundary of the gray region that all globally minimize worst-case risk (on the red task). But L-DRO systematically converges to the optimal point at the intersection of the x-axis and the gray region’s boundary.
>
>
> > Additional baseline
>
> This is an interesting suggestion. Note that an issue with this baseline is that in general, it may not achieve the optimal worst-case solution, but rather will interpolate between DRO and static mixing. L-DRO on the other hand is designed to look for solutions which have the same worst-case performance as DRO, but achieve higher average task performance. We tried this experiment on CIFAR-100 (averaging the two losses) and find that this baseline is generally outperformed by L-DRO. We refer to the updated paper for the details of this additional experiment.
>
> > Performance of baselined-DRO
>
> Note that in the more realistic setting of our LM experiment, we are operating with a more efficient version of L-DRO which is the result of many simplifying approximations, which explain that its performance is not as univocally better than expected in theory. Still, we find that it performs reasonably, especially compared with DRO.
>
> We agree that baselined-DRO’s performance is a good argument for DRO-like approaches, despite its possible disadvantages in terms of computational overhead, training and storage of additional models, etc... We will adjust our introduction to better reflect this fact.
>
> > Computational overhead of L-DRO vs baselined DRO
>
> We discuss the wall-clock time of L-DRO in more detail in our response to all reviewers. In summary, we find that L-DRO is approximately 1.55 times slower than ERM or DRO in our LM experiments. In comparison, Baselined-DRO necessitates training 49 additional monolingual models. It is true that baselined-DRO’s cost could be reduced by reducing the size of the baseline monolingual models, but it is unclear how the resulting degradation in the estimate of the baseline loss might impact the performance of the algorithm.

---

### Official Review · Reviewer_NzDd · 2021-11-03

**Correctness:** 3
**Technical Novelty And Significance:** 3
**Empirical Novelty And Significance:** 3
**Recommendation:** 5
**Confidence:** 4

**Details Of Ethics Concerns:**

None.

**Main Review:**

Overall, I think the submission is novel and addresses an important question in multi-task learning. I believe the algorithm has a clear theoretical motivation but also want to note that the author makes a key simplification (i.e. the first order approximation) in estimating the task interaction, which most likely does not hold for non-convex neural network optimization. My main concerns are with the empirical evaluation of the proposed method, which are detailed below. Currently, my stand for the submission is "marginally below the acceptance threshold" but I can see the potential of the proposed method and could be convinced otherwise if the authors can help answer my questions/requests.

I have the following list of questions:

1. In practice, the loss values from different tasks can be very different. For example, if we learn a neural network to perform image recognition and segmentation simultaneously, the loss values will be of different scales from these two tasks. What would the authors recommend for such use case?
2. The authors propose to renormalize the gradients into one when estimating the "interaction matrix" but provides little intuition and empirical validation so I am not convinced that this is a principled technique. Normalizing the gradients breaks the analysis section 2.5 which creates a gap between the conceptual motivation and the actual algorithm, which is quite disappointing. Can the authors justify this decision? Also, I would like to see ablation study on this design decision.
3. The authors propose to reduce the computational cost by estimating the interaction matrix periodically. Can I see an ablation study on this hyper-parameter to understand the trade-off here? How much performance do we lose when we set the estimation period to be 1000 steps?
4. The CIFAR100 experiment in the appendix (Figure 5) is quite concerning. I appreciate the honest documentation of this experiment. I wonder if this is because weighting approach in general does not work with overparametrized networks [1]? Can the authors provide more insights on this experiment?
5. Can the authors document the actual wall clock time and memory overhead of the proposed method on their CIFAR and Multilingual LM experiments?
6. For Table 1, the difference between absolute perplexity and relative perplexity is quite confusing. We observe that DRO achieves the best worst case absolute perplexity but not so much for relative perplexity. I am not entirely sure the "relative perplexity" is the correct metric: perplexity is defined as the inverse of the average probability of predicting the right word so computing the difference and the division does not have a clear definition. I understand the difficult in comparing perplexities across languages. Can the authors provide a table of the full list of perplexity values on different languages in the Appendix so I can better judge the results?
Aside from my current confusion mentioned above, the empirical gain of L-DRO over uniform sampling (static mixing with alpha=0) is quite marginal, which is concerning.

**Summary Of The Paper:**

This paper studies balancing average case and worst case performance in a multi-task setting. The authors propose a new algorithm that extends the group DRO algorithm by 1. ) taking task interactions into account 2.) enforcing the maximum entropy principle when breaking ties. The interactions between tasks are modeled by a first order approximation to change in training loss after a single gradient update. Empirically, the authors propose methods to stabilize their estimation of the "task interaction" estimation by a online weight update procedure and also introduce an additional technique that re-normalize gradients to unit norm before calculating the "task interaction" estimation. In addition, the authors observed that the proposed algorithm only needs to update the task interaction estimation periodically during training, which reduces the computational overhead.

The submission empirically test the proposed algorithm on CIFAR100 and Multilingual Language Modeling. On CIFAR100, the proposed method achieves improved performance for smaller-scale neural networks but does not achieve significant improvement on larger neural networks. On Multilingual Language Modeling, the proposed method achieves best average performance and second best worst case performance (trailing a computationally expensive baseline).

**Summary Of The Review:**

Novel and interesting algorithm that studies a important problem in multi-task learning. Current empirical evaluation of the paper can be furthered improved. Several design decisions in the final version of the algorithm should be better justified.

---

> ### Author Response · Authors · 2021-11-22
> **Response to Reviewer NzDd**
>
> We thank the reviewer for feedback and their detailed list of questions. We respond to specific points below:
>
> > What happens when the various losses have different scales
>
> To some extent, we think our experiments (especially the synthetic experiments in Section 4, but also the multilingual LM experiments) show that L-DRO is able to handle tasks with varying loss scales (given the difference in baseline performance across languages). However, exploring more extreme scenarios with drastic differences in scale across tasks would certainly be interesting and we leave it for future work.
>
> > Importance of periodic evaluation of the interaction matrix
>
> Our implicit assumption with this approximation is that the interaction between tasks (as measured by pairwise dot products) changes at a slower rate than the actual loss values of each task, which is why we can afford to estimate the former only periodically, whereas the latter is recomputed at every step.
>
> In practice, recomputing the interaction matrix less frequently comes at the cost of increased instability, especially for high values of the lookahead learning rate: in these cases using a stale interaction matrix can lead to a divergence of the weights to a unimodal distribution from which it can be hard to recover. We observe this fact in an ablation study on CIFAR-100 with three different values of the update interval (10, 100, 1000). We find that training curves for 10 and 100 are similar, but the larger interval (1000) leads to instability. We include this ablation in the revised version of the paper (in the appendix).
>
> > Gradient renormalization breaks the theoretical interpretation of L-DRO
>
> Note that there of the two components in each element in the interaction matrix $A_{ij}=\nabla\ell_{i}^\intercal\nabla\ell_j$ and we only normalize one ($\nabla\ell_{i}^\intercal\nabla\ell_j\rightarrow \nabla\ell_{i}^\intercal\frac{\nabla\ell_j}{\vert\nabla\ell_j\vert}$).
>
> This can be interpreted as the first order approximation of taking a lookahead step along the *normalized* task gradients:
>
> $$\ell_i(\theta_0-\lambda[\sum_{j=1}^N w_j\frac{\nabla\ell_{j}(\theta_0)}{\Vert \nabla\ell_{j}(\theta_0)\Vert}])
> \approx\ell_i(\theta_0) - \lambda \sum_{j=1}^Nw_j\nabla\ell_{i}(\theta_0)^\intercal\frac{\nabla\ell_{j}(\theta_0)}{\Vert \nabla\ell_{j}(\theta_0)\Vert}$$
>
> Our intuition with this renormalization trick is that it helps mitigate the instability caused by the “staleness” of the interaction matrix with periodic updates. Indeed, the approximation error of using a stale interaction matrix comes from two sources: outdated alignment between the gradient (cosine similarity) and outdated gradient norm. With our renormalization trick, we mitigate the latter by factoring out the norm of the lookahead step while keeping the original interpretation (as explained above).
>
> In practice, we do observe that this helps improve stability for larger matrix update interval. Indeed, in an ablation study in CIFAR-100, we find that the use of the renormalization trick removes the instability observed with larger update intervals (1000). Again, we added this ablation to the paper (along with the discussion above).
>
>
> >Can the authors document the actual wall clock time and memory overhead of the proposed method on their CIFAR and Multilingual LM experiments?
>
> Please see our response to all reviewers where we discuss this. We added this information in the revision’s appendix
>
> > Interpretation of relative perplexity
>
> There is actually a natural interpretation of the relative perplexity. Indeed, the perplexity of a model can be interpreted as the exponential of its cross-entropy with the underlying data distribution, ie $\text{Perplexity}(\theta, i)=e^{H(p_i,p_\theta)}$ where $p_\theta$ is the model’s distribution and $p_i$ is the data distribution corresponding to language $i$. We can thus rewrite the relative perplexity as
>
> $$\text{Relative perplexity}(\theta, i)=\frac{\text{Perplexity}(\theta, i) - \text{Perplexity}(\theta^*_i,i)}{\\text{Perplexity}(\\theta^*_i, i)}=\frac{\text{Perplexity}(\theta, i)}{\text{Perplexity}(\theta^*_i, i)} -1=e^{H(p\_i, p\_\theta) - H(p\_i,p\_{\theta^*\_i})} -1$$
>
>
> Note that we can go one step further if we assume that $\\theta^{\*}\_{i}$ is globally optimal (a perfect monolingual model), then the second term approximates the entropy $ H(p\_i,p\_{\theta^*\_i}) \approx  H(p\_i)$, and the relative entropy can be interpreted as the exponentiated KL divergence between $p\_\theta$ and $p\_i$ (up to the constant $-1$ term):
>
> $$
> \text{Relative perplexity}(\theta, i)\approx e^{KL(p\_i || p\_\theta)} -1
> $$
>
> > CIFAR-100 results
>
> This was a common question among reviewers, hence we refer to our general response for a discussion.

---

### Author Response · Authors · 2021-11-22
**General Response and Paper Revision**

We thank the reviewers for their in-depth comments and their helpful feedback.

We just updated a revised version of the paper including clarifications and additional results. The core changes to this version are:

- A more detailed discussion of the computational overhead of L-DRO, including effective runtime for our implementations (compared to the runtime of static mixing), in the appendix
- Clarifications in the method section (especially with regards to the introduction of L-DRO).
- Ablation studies on the interaction matrix update interval and the renormalization step (in the appendix)
- Results for an additional baseline on CIFAR-100: interpolation between average and worst-case loss (in the appendix)

Below, we address salient points that were brought up by multiple reviewers. We respond to each reviewer independently in separate comments.


> Overhead (time/memory) of L-DRO

As mentioned in the paper, the main overhead of L-DRO is the computation of the Lookahead matrix. The fastest way of doing so is to compute the $n$ task gradients and take their pairwise dot-product. The overhead incurred is equivalent to taking $n$ forward/backward passes (one to obtain each of the task gradients), as the cost of the pairwise dot-product is minimal in comparison. In theory, this means training with L-DRO becomes approximately $n/T$ times slower, where $T$ is the lookahead matrix update frequency.

We employed this strategy for the CIFAR experiments. With $n=20$ and $T=100$ in that setting, the theoretical slowdown is approximately $\times 1.2$ . In practice, for our biggest CNN model on CIFAR-100, training speed was around 27 steps/s for L-DRO vs 28.5 steps/s for ERM (for reference, PCGrad ran at 21 steps/s).

The downside of this approach is that it necessitates holding $n$ gradients in device memory. This can quickly become cumbersome when the number of tasks and the size of the model are large. For the language modeling experiment, we adopt the alternate---more memory efficient---strategy of re-computing each dot-product separately. This means the memory overhead drops to only holding $2$ gradients in memory at each time, but we need to compute $O(n^2)$ backward passes. In practice for the language modeling experiment, we had $n=49$ and $T=1000$, leading to a theoretical slowdown of approximately $\times 3.4$. By optimizing the interaction matrix computation (eg. computing only the upper triangle, using a smaller (2x smaller) batch size to estimate the gradients), we end up with a training speed of 2.2 steps/s for L-DRO vs approximately 3.4 steps/s for ERM, i.e. an effective slowdown of $\times 1.55$.

Note that another way to reduce device memory usage would be to use the first approach and offload the computation of the gradient dot-products to the CPU (where memory is much less of an issue), in which case the additional overhead comes from the cost of moving gradients from device to cpu as well as computing large dot-products on cpu. However, we did not experiment with this last approach.


> Performance on CIFAR with large models

A few reviewers asked about our interpretation of the results in the Appendix showing that all methods performed similarly on CIFAR-100 when the model is sufficiently large. We agree with reviewer NzDd’s suggestion that this might be an instance of weighting approaches not working on over-parameterized deep learning models (eg. Byrd & Lipton, 2018; https://arxiv.org/abs/1812.03372).

That being said, we would like to emphasize that the multitask-CIFAR benchmark is very small compared to realistic datasets, and as such we would advise against making pessimistic conclusions based on these results. Indeed, as evidenced in our language modeling experiments, we do not observe this phenomenon in practice, despite using a large model.

---

### Decision · Program_Chairs · 2022-01-20

**Decision:**

Reject

**Comment:**

In general, the reviewers recognized the importance of the question and the innovation in the proposed algorithm, but do not seem to be super excited about the overall contribution of the paper. (One or two reviewers did not seem to respond authors' response after the AC's reminder.) The AC read the reviews and responses and observed that the main concern appears to be the empirical performance --- the improvements are not as strong for larger models or if more computational time is allowed. Modern models are indeed typically large, and it would be good to discuss this point more thoroughly. If the work's focus is limited resource setting, the paper might want to state that upfront. Indeed, one reviewer is still concerned post-rebuttal about a clock-time comparison. Given these considerations, the AC will recommend reject for the paper but encourage the authors to resubmit to a top venue conference after revising the paper.